# Efficacy of Sildenafil in Infants with Bronchopulmonary Dysplasia-Associated Pulmonary Hypertension

**DOI:** 10.3390/children10081397

**Published:** 2023-08-16

**Authors:** Kacie Dillon, Vineet Lamba, Ranjit R. Philip, Mark F. Weems, Ajay J. Talati

**Affiliations:** 1Division of Neonatal-Perinatal Medicine, Department of Pediatrics, University of Tennessee Health Science Center, Memphis, TN 38163, USA; 2Division of Pediatric Cardiology, Department of Pediatrics, University of Tennessee Health Science Center, Memphis, TN 38163, USA; rphilip@uthsc.edu

**Keywords:** bronchopulmonary dysplasia, BPD, pulmonary hypertension, sildenafil

## Abstract

**Background:** Pulmonary hypertension (PH) is a common comorbidity in infants with bronchopulmonary dysplasia (BPD). Sildenafil is a widely recognized therapy for PH, but its efficacy in infants with BPD is questionable. We propose to assess the efficacy of sildenafil in BPD-associated PH as evaluated based on transthoracic echocardiography (TTE) changes and clinical measures. **Methods:** Data were retrospectively and prospectively collected. Inclusion criteria were gestational age (GA) < 32 weeks, birth weight (BW) < 1500 g with severe BPD, diagnosis of PH via TTE on sildenafil treatment. PH was evaluated via TTE, which was performed monthly after 36 weeks post-menstrual age (PMA) as a standard of care, and re-reviewed by a single pediatric cardiologist, who was blind to the initial reading. **Results:** In total, 19 patients were enrolled in the study, having a median GA of 24 3/7 weeks (IQR 23 5/7–25 5/7) and a median BW of 598 g (IQR 572–735). Sildenafil treatment was started at a median PMA of 40.4 weeks. The median respiratory severity score (RSS) at 28 d was 6.5, RSS and FiO2 showed improvement about 12 weeks after starting sildenafil treatment. **Conclusions:** Improvement in PH was noted via TTE, and patients had improvement in their RSS and FiO2 after prolonged therapy. However, TTE improvements did not correlate with clinical improvements.

## 1. Introduction

Due to numerous advancements in the field of neonatology, survival of extremely premature and extremely low birth weight infants has increased; however, these infants often suffer from impaired lung development and require long-term ventilator support and/or supplemental oxygen, putting them at risk of developing bronchopulmonary dysplasia (BPD) [1,2,3]. In the past 20 years, BPD incidence has remained at about 40% in surviving infants born at 28 weeks of gestation or less, indicating that more progress must be made to reduce BPD cases [2]. In infants with BPD, both lung alveolarization and angiogenesis are disturbed [2,3]. This disruption in angiogenesis can result in increased pulmonary vascular resistance and elevated pulmonary pressure, leading to right ventricular hypertrophy and pulmonary hypertension (PH) diagnosis [2]. Development of PH in BPD has been shown to negatively affect the infant and increase mortality [2].

Frequent evaluation to identify PH in patients with BPD using routine transthoracic echocardiogram (TTE) surveillance is now the standard of care based on recommendations from the Pediatric Pulmonary Hypertension Network (PPHNet) [4]. According to recommendations outlined by the PPHNet, “infants with evidence of significantly elevated pulmonary vascular resistance and right ventricular impairment (moderate hypertrophy or dysfunction) not related to left heart disease or pulmonary vein stenosis” should be started on pharmacologic therapy [4]. As a result, use of pharmacotherapy to treat BPD-associated PH is steadily increasing; however, not much is known about long-term efficacy and safety of many of these drugs, and with the exception of nitric oxide, none of these drugs have been approved by the FDA for the treatment of infants [2]. One such drug is a phosphodiesterase inhibitor, namely sildenafil. By selectively inhibiting phosphodiesterase type 5 (PDE-5), sildenafil increases the amount of circulating cyclic guanosine monophosphate (cGMP), which, in turn, increases vasodilation and inhibits smooth muscle growth [1,5].

In our neonatal intensive care unit (NICU), sildenafil is frequently used as the first-line choice of treatment for PH based on TTE findings. While there are limited data regarding the use of sildenafil in BPD-associated PH, a meta-analysis reviewing several small retrospective studies implied a reduction in pulmonary arterial pressure and improvement in respiratory scores occur through the use of sildenafil [5]. Regarding adverse effects, it is theorized that sildenafil may worsen retinopathy of prematurity (ROP) by enhancing angiogenesis; while a few small studies did not show a significant increase in ROP in infants treated with sildenafil, more research needs to be conducted to investigate this tentative finding [6,7,8].

The goal of our study was to determine the efficacy of sildenafil use in infants with BPD-associated PH through prospective and retrospective studies, considering improvement in TTE findings to be our primary outcome. We hypothesized that sildenafil use will improve pulmonary hypertension in infants with severe BPD. Additionally, we evaluated supplemental oxygen requirements and respiratory severity scores (RSS) before and after sildenafil use and evaluated potential side effects of sildenafil, specifically ROP. 

## 2. Materials and Methods

Data were prospectively and retrospectively obtained regarding a cohort of infants with severe BPD-associated PH who were being treated with sildenafil. Retrospective data were collected regarding patients admitted to our level 3 and level 4 NICUs between January 2018 and November 2020. Participants in our NICUs between December 2020 and January 2022 were included in the prospective cohort. Inclusion criteria were gestational age (GA) of <32 weeks, birth weight <1500 g, severe BPD, diagnosis of PH via TTE, and receiving sildenafil treatment. Exclusion criteria were infants with congenital heart disease (except patent ductus arteriosus (PDA), a patent foramen ovale/atrial septal defect <5 mm or a ventricular septal defect (VSD) <2 mm) and lethal congenital abnormality. Each patient served as their own control, as the severity of PH and clinical illness was compared over time. 

BPD was defined according to NIH guidelines as follows: a patient born prior to 32 weeks who, at 28 days of age, still required respiratory support to maintain oxygen saturation >90%. Moderate BPD was defined as the need for <30% FiO2 at 36 weeks post-menstrual age (PMA), and severe BPD was defined as the need for ≥30% O_2_ or nasal CPAP/HFNC >2 LPM at ≥36 week PMA [9]. For infants receiving nasal cannula ≤2 LPM, effective O_2_ delivery was calculated using a table from the 2005 Walsh et al. article published in Pediatrics [10]. This table takes into account current weight, flow, and FiO2 in order to determine the effective FiO2 [10]. 

Using recommendations from the “Evaluation and Management of Pulmonary Hypertension in Children with Bronchopulmonary Dysplasia”, which was developed by the PPHNet, TTEs were performed at 36 weeks PMA (the age of formal BPD diagnosis) as a standard care [4]. As part of this study, TTEs were re-reviewed by a pediatric cardiologist, who was blind to the original report. They determined PH diagnosis based on interventricular septum position in systole, as well as other parameters, such as the peak velocity of tricuspid regurgitation, and gradients across the PDA or small VSD were used when available for interpretation. The secondary effects of PH, i.e., right ventricular hypertrophy (RVH) and dilation were also assessed. The PPHNet paper defined significant TTE findings as “estimated RVSP (right ventricular systolic pressure)/LVSP (left ventricular systolic pressure) >0.5 and septal flattening in the absence of a significant left to right shunt” [4]. Based on this definition, these patients with moderate-to-severe PH should be initiated on sildenafil or another pharmacologic agent. Based on the readings generated by our patient-blind cardiologist, moderate PH was defined as an estimated RVSP of 1/2–2/3 systemic pressure with septum flattening and moderate RVH or dilation [4]. Anything worse than this standard was classified as severe [4]. The clinical team decided to use pharmacotherapy, and we enrolled patients in the study once sildenafil treatment was initiated. 

We recorded the age of initiation of sildenafil therapy, the starting dose, and any dose adjustments, as well as any other pulmonary hypertension medications used in conjunction. After treatment with a pharmacologic agent was started, TTEs were usually repeated every 2–4 weeks using standard care guidelines for BPD/PH management.

The primary goal of the study was to assess improvement in TTE findings of PH after the initiation of sildenafil treatment. Specifically, we tested for a decrease in intraventricular septal flattening and a decrease in RVSP. The secondary goals included any improvement in FiO2 requirements and respiratory severity scores (mean airway pressure × FiO2) after starting sildenafil treatment. We also evaluated the effect on ROP, which is thought to be altered by the potential angiogenesis effect of sildenafil and is a theorized adverse effect. 

Infants were enrolled via chart review. Study data were collected and managed using the REDCap electronic data capture tools database [11,12]. Deidentified data were then exported to a Microsoft Office Excel 2010 spreadsheet. Statistical analysis was performed using formulas derived from Microsoft Office Excel 2010. Medians were used to describe baseline characteristics, as well as the characteristics at each TTE. 

## 3. Results

Twenty-nine infants with severe BPD were screened from January 2018 to January 2022, of which nineteen infants were enrolled in the study (Figure 1). Of these nineteen infants, ten infants were in the prospective group and nine infants were in the retrospective group. The median GA was 24 3/7 weeks (IQR 23 5/7–25 5/7), and the median birth weight was 598 g (IQR 572–735). The majority of infants (63%) were mechanically ventilated. Table 1 outlines full baseline characteristics and demographic information related to the enrolled infants. 

Table 2 shows the severity of PH on serial TTEs and its correlation with PMA, respiratory severity score (RSS), FiO2, pCO_2_ (from blood gas), and sildenafil dose at each TTE. Sildenafil was usually initiated after the first TTE. The median PMA at first TTE was 40.3 weeks, and the median PMA at which sildenafil was started was 40.1 weeks. Approximately half of the cohort had moderate or severe PH on their first TTE. The median dose of sildenafil was 2.4 mg/kg/day. By the fourth TTE, the number of infants with moderate/severe PH dropped to 30% (*n* = 6) of the entire cohort and, similarly, 30% (*n* = 6) of the infants had no PH. A significant proportion of mild PH seemed to be resolved by the second TTE.

Figure 2 demonstrates the changes in FiO2 and RSS with increasing PMA following sildenafil treatment initiation. The RSS and FiO2 were noted to trend toward improvement about 14 weeks following the initiation of sildenafil treatment. Figure 3 shows the correlation between RSS and the degree of PH. No correlation was noted, in other words, and a high RSS did not always correlate with a higher severity of PH via TTE, and vice versa. 

Table 3 contains information regarding the NICU course of the 19 study participants. In total, 4 of the 19 participants died prior to NICU discharge, and 10 patients received a tracheostomy during their NICU stays. The median length of stay was 369 days (IQR 248–522 days). Regarding adverse effects, ROP data were recorded. In total, 3 of the 19 patients were noted to have Stage 3 ROP in our cohort. Of those individuals, 1 patient was noted to have Stage 3 ROP prior to sildenafil treatment initiation. All 3 patients were treated via intravitreal bevacizumab injection and/or laser surgery for retinopathy of prematurity. Of these 3 participants, 1 patient had both bevacizumab and laser surgery performed prior to sildenafil treatment initiation. Finally, 12 participants were fully vascularized or had no ROP prior to starting sildenafil treatment.

## 4. Discussion

Our study represents a cohort of patients with severe BPD and PH. Despite previously reports, we believe that our study is the first study comparing TTE findings and clinical correlation using RSS following sildenafil treatment initiation. We also believe this to be the first study with a prospective component investigating sildenafil use in infants with a diagnosis of severe BPD and PH. We found improvement in mild PH on TTE and a greater number of patients with no PH after 14 weeks of sildenafil therapy. We also found improvement in clinical outcomes at 14 weeks following sildenafil treatment initiation. However, since there is no control arm in the study of infants who were not exposed to sildenafil and as BPD is a multifactorial disease, it is unclear if use of sildenafil contributed to these clinical improvements. Additionally, improvement in the RSS did not always correlate with improvement in PH via TTE. However, it is notable that patients with no PH were more likely to have lower RSS, and the highest RSSs were found in those with severe PH. There was no significant early improvement in moderate/severe PH after initiation of sildenafil treatment. However, the right ventricular function was preserved in this cohort, possibly suggesting adequate afterload reduction in the right ventricle after sildenafil treatment initiation. 

A meta-analysis from 2018 reviewed five small retrospective studies regarding the use of sildenafil in infants with BPD and PH [5]. The pooled mean EGA in this meta-analysis was slightly older than that of our cohort at 26.3 weeks +/− 2.5. Sildenafil treatment was started at an older age in most of the studies using a pooled mean post-natal age at sildenafil treatment initiation of 48.2 weeks +/− 15.9 [5]. Additionally, the maximum dose of sildenafil in these studies ranged from 6 to 8 mg/kg/day [5], whereas the maximum dose in our study was 4 mg/kg/day. However, much of the data included in this analysis were recorded in patients treated with sildenafil prior to the FDA warning recommending against high-dose sildenafil treatment, which was issued on 30 August 2012 [13]

This meta-analysis reviewed improvements in pulmonary arterial pressure (PAP) and respiratory scores (FiO2 and RSS) following sildenafil treatment initiation. These areas could not be assessed in two of five of the studies due to the fact that individual participants’ parameters were not separately pooled [5]. Two of the studies used TTE findings to infer improvement in PAP, which was similar to the approach used in our study, and one study used catheterization to determine PAP, which is the gold standard; however, it is more difficult to perform given its invasive nature compared to TTE. The researchers found a 69.3% improvement in estimated PAP and a 15% improvement in respiratory scores following sildenafil treatment initiation [5]. 

While some studies in this meta-analysis reported both respiratory severity scores and TTE or catheterization findings following sildenafil treatment initiation, these data points were recorded at different time points [14,15], unlike our study, in which RSS and echocardiogram results were recorded on the same days. One such study performed by Trottier-Boucher et al. reviewed time to clinical improvement using RSS and vent requirements and improvement in PAP using TTE findings [14]. This study was similar to ours in that the median EGA was 26 weeks and the median PMA of sildenafil treatment initiation was 40 weeks; however, the median dose of sildenafil was much higher than that of our study, being at a dose of 4.4 mg/kg/day, with the maximum dose being 7.3 mg/kg/day [12]. This case study reported clinical improvements in 35% of 23 patients following sildenafil treatment initiation, with 75% of these improvements occurring at 48 h following initiation, with no further report of clinical outcomes following this time period existing [14] Improvement in PH on TTE was reported in 71% of 21 patients, as 2 patients had incomplete TTE data [14]. A similar study was Nyp et al. found significantly reduced RVSP following sildenafil treatment initiation; however, it did not find similar improvements in RSS after reviewing scores up to only 48 h after sildenafil treatment initiation [15].

Kadmon et al. was another study included in the analysis, which measured clinical improvement after sildenafil treatment initiation using the Modified Ross Heart Failure Classification for Children rather than RSS [16]. The authors also used TTE data, as well as cardiac catheterization data, to evaluate PH improvement. They followed their cohort for 24 months and found a significant improvement in TTE findings in 14 of 18 patients treated with sildenafil [16]. 

Regarding other studies included in the meta-analysis, Tan et al. reviewed a cohort of 22 severe BPD–PH patients reporting data at the initiation of sildenafil treatment and 4 weeks after sildenafil treatment began [17]. They found significant improvements in TTE findings following initiation of sildenafil treatment; however, respiratory parameters measured using FiO2 requirements and PCO_2_ within 1 week of echocardiogram were not significantly improved [17]. This outcome was similar to our study, in which TTE data did not always correlate with clinical status. Lastly, Mourani et al. found gradual improvements in PH in a cohort of 25 infants with chronic lung disease (18 with BPD) over a 2-year period, but they did not record respiratory clinical scores or FiO2 needs [18].

Regarding adverse effects, we found only two patients with worsening of ROP after sildenafil treatment initiation. This result was similar to those of the studies included in the meta-analysis, which reviewed ROP stages before and after sildenafil treatment initiation [14,17,18]. One study found no statistically significant difference in the degree of ROP before and after sildenafil treatment initiation [14], one study found no progression in ROP after sildenafil treatment initiation [17], and another study stated that the data that they recorded regarding ROP did not suggest a worsening in disease after starting sildenafil treatment [18]. 

Some limitations of our study included its small sample size, lack of a control group, and pulmonary hypertension being diagnosed using TTE rather than the gold standard method of cardiac catheterization. We attempted to compare our cohort to another cohort of patients diagnosed with PH via TTE who were not started on sildenafil treatment; however, this group had a significantly lower baseline RSS and were not comparable to our cohort. Regarding TTE findings, 1 patient was found to have no PH, and 10 patients were found to only have mild PH prior to the start of sildenafil treatment when reviewed by our patient-blind pediatric cardiologist. This finding may be due to the original echocardiogram reader defining a more severe degree of PH or the degree of clinical severity. 

The strength of this study is that we were able to include a prospective component for up to 4 months after starting sildenafil treatment, which has not been previously reported in this cohort of patients. Additionally, we were able to compare TTE findings to the clinical outcomes of RSS and FiO2 over a 3-month period, which has also not been previously reported. 

More research needs to be conducted regarding the effectiveness and dosing of sildenafil treatment. Our study complied with the institutional protocol of sildenafil dosing, i.e., between 2 and 4 mg/kg/day. Ultimately, a multi-center prospective randomized trial would be beneficial in providing more data regarding the use of sildenafil, but this outcome may not be feasible in all patients, as treatment for moderate-to-severe PH is currently the standard of care [4].

## 5. Conclusions

Clinical improvement (decreased FiO2 and RSS) occurred about 14 weeks after sildenafil treatment initiation. There was a decrease in mild PH determined via TTE after 4 weeks of sildenafil treatment and an increase in babies with no PH after 4 weeks; however, moderate/severe PH did not show early improvement. This finding suggests that Sildenafil therapy can be expected to have a prolonged duration prior to showing any TTE or clinical improvement. The severity of PH on TTE did not correlate with clinical status. This result may indicate that sildenafil may make a fractional contribution to clinical improvement in BPD-associated PH. However, BPD is a multifactorial disease, and the improvement noted in our patients cannot be solely attributed to Sildenafil, as it is possible that the ventilator strategy and other management decisions more notably account for clinical improvement. The role of sildenafil and other pulmonary vasodilators remains ill-defined, and further studies are needed.

## Figures and Tables

**Figure 1 children-10-01397-f001:**
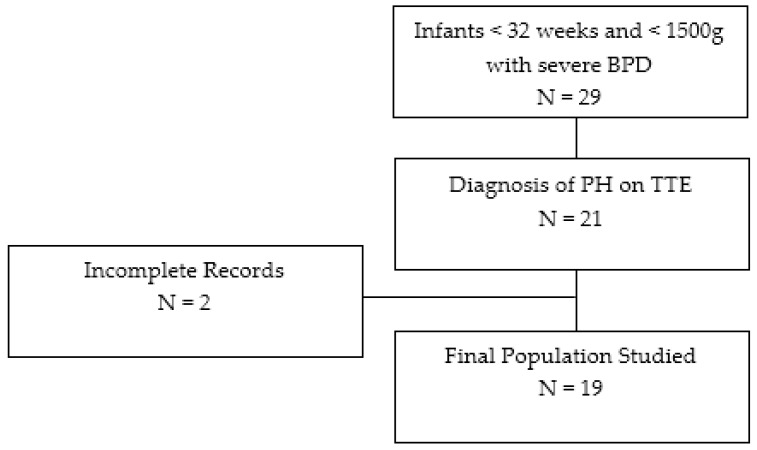
Flowchart of study population selection.

**Figure 2 children-10-01397-f002:**
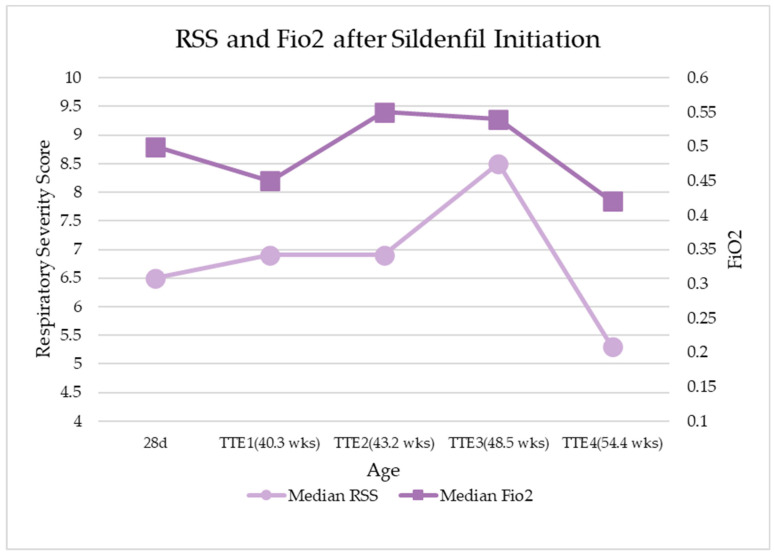
Comparisons between RSS and FiO2 at different CGAs prior to and following sildenafil treatment initiation.

**Figure 3 children-10-01397-f003:**
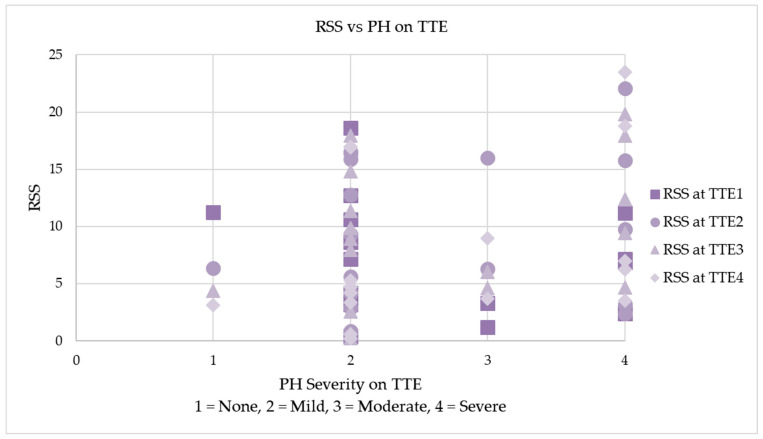
Comparison between RSS at differing degrees of PH. PH is described numerically, with 1 = no PH, 2 = mild PH, 3 = moderate PH, and 4 = severe PH.

**Table 1 children-10-01397-t001:** Patient characteristics.

Baseline Characteristics	
*n*	19
Median EGA, completed weeks (IQR)	24 3/7 (23 5/7–25 5/7)
Median BW, grams (IQR)	598 (572–735)
Sex	
Male, *n* (%)	13 (68)
Female, *n* (%)	6 (32)
Race	
Black, *n* (%)	17 (89)
White, *n* (%)	2 (11)
Respiratory support at 36 weeks PMA	
IMV/HFV	12 (63)
CPAP/NIPPV	4 (21)
HFNC (>2 LPM, >30% FiO2)	3 (16)

EGA = estimated gestational age. IQR = interquartile range. BW = birthweight. PMA = post-menstrual age. IMV = intermittent mandatory ventilation. HFV = high-frequency ventilation. CPAP = continuous positive airway pressure. NIPPV = non-invasive positive pressure ventilation. HFNC = high-flow nasal cannula. LPM = liters per minute. FiO2 = fraction of inspired oxygen.

**Table 2 children-10-01397-t002:** Clinical severity and PH severity via TTE.

	TTE 1	TTE 2	TTE 3	TTE 4
Median PMA, Weeks	40.3	43.2	48.5	54.4
Median RSS	6.9	6.9	8.4	5.3
Median FiO2	0.45	0.55	0.535	0.42
Median pCO_2_	59	60	60.5	55
Sildenafil Dose, Median (range), mg/kg/day	0	2.4 (1.2–4)	2.5 (1–4)	3 (1–4)
PH Severity on TTE, *n*				
None	1	7	6	6
Mild	10	5	6	5
Moderate	4	4	4	3
Severe	5	3	2	3

TTE = transthoracic Echo, PMA = post-menstrual age, RSS = respiratory severity score, FiO2 = fraction of inspired oxygen, PCO_2_ = partial pressure of carbon dioxide PH = pulmonary hypertension.

**Table 3 children-10-01397-t003:** Characteristics of NICU course.

NICU Course Outcomes	
*n*	19
PDA Device Closure, *n* (%)	7 (37)
Post-Natal Steroids, *n* (%)	19 (100)
Bosentan, *n* (%)	1 (5)
Death, *n* (%)	4 (21)
Tracheostomy, *n* (%)	10 (53)
IVH Grade 3–4, *n* (%)	5 (26)
NEC Stage 2B or Greater, *n* (%)	3 (16)
Length of stay, Median Days (IQR)	369 (248–522)

PDA = patent ductus arteriosus. IVH = intraventricular hemorrhage. NEC = necrotizing enterocolitis.

## Data Availability

The data presented in this study are available on request from the corresponding author.

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
