# Peer review of "Efficacy of Sildenafil in Infants with Bronchopulmonary Dysplasia-Associated Pulmonary Hypertension"

_children, 2023, doi:10.3390/children10081397_

Round 1

Reviewer 1 Report

This is a small study that provides invaluable information from BPD-associated PH patients.

Abstract: Please define "RSS"

Figure 1: Please provide statistical information.

Author Response

This is a small study that provides invaluable information from BPD-associated PH patients.

Abstract: Please define "RSS"
Corrected

Figure 1: Please provide statistical information.

We are unable to perform meaningful statistics, as the cohort is very small. We are just reporting data trends over time. This has been clarified in the text.

Reviewer 2 Report

Efficacy of Sildenafil in Infants with Bronchopulmonary Dyplasia associated Pulmonary Hypertension.

In this study the authors aimed to assess the efficacy of sildenafil in BPD associated PH as evaluated by transthoracic echocardiography (TTE) changes and clinical measures in infants <32 weeks GA and <1500 g BW, with severe BPD and PH on sildenafil treatment. They performed monthly TTE from 36 weeks on, and their main finding was improvement in PH on TTEs, and patients improvement in their RSS and FiO2 after a prolonged therapy. However, there was not correlation between these improvements, probably due to the multifactorial origin of the disease.

Major issues.

The study is well written. But, from a methodological point of view, the study has some important shortcomings. As recognized in the limitations section, the sample is small and, above all, there is no control group, so it is not possible to determine if the evolution presented by the patients is due to the intervention (sildenafil) or is simply the natural evolution of the disease.

For an adequate methodology concerning cohort studies, please take into account the recommendations that can be found in https://www.strobe-statement.org/checklists/

The checklist that accompanies this information is very useful for an adequate presentation of this type of studies.

Minor issues.

Abstract.

Typo. Line 12. Propsed.

Lines 18-19. Please, consider giving IQR apart from medians for GA and BW, if possible. This is more informative.

Please, explain the abbreviation RSS the first time it appears (line 19).

Once it has been explained, use it consistently. For instance: gestational age (GA) in line 14, use “GA” in line 70 instead of “gestational age”, line 117-118, etc.

Material and Methods.

Line 97. Typo: “severe [4]/ The…”

Were all the infants who fulfilled the inclusion criteria during the study period included? A flowchart showing the eligible patients and the exclusions with the reason for it would be very interesting to see.

Results.

Line 128-9. Please avoid “postmenstrual gestational age”.  PMA has been already explained in the abstract and in line 74. It is not necessary to repeat it again. Use just PMA.

The same for RSS. It is not necessary to define it again.

Figure 2. Please, explain what each of the symbols mean (circles, squares, crosses, etc.).

Discussion.

Line 181. It reads “at 26.3 weeks +/- 2.7.5 Sildenafil was started…” Please, correct it.

In this paragraph, reference [12] is cited twice (lines 204 and 208), but probably you meant [14]. Please, double check.

Author Response

Major issues.

The study is well written. But, from a methodological point of view, the study has some important shortcomings. As recognized in the limitations section, the sample is small and, above all, there is no control group, so it is not possible to determine if the evolution presented by the patients is due to the intervention (sildenafil) or is simply the natural evolution of the disease. For an adequate methodology concerning cohort studies, please take into account the recommendations that can be found in https://www.strobe-statement.org/checklists/
The checklist that accompanies this information is very useful for an adequate presentation of this type of studies.

We have incorporated some of the guidelines in this checklist and clarified that the control in our cohort is each patient as we are studying the evolution of their disease process over time.

Minor issues.

Abstract.

Typo. Line 12. Propsed.-

corrected

Lines 18-19. Please, consider giving IQR apart from medians for GA and BW, if possible. This is more informative.

We have added IQR to abstract and changed results table and text to include that rather than range.

Please, explain the abbreviation RSS the first time it appears (line 19).

Corrected

Once it has been explained, use it consistently. For instance: gestational age (GA) in line 14, use “GA” in line 70 instead of “gestational age”, line 117-118, etc.

Corrected

Material and Methods.

Line 97. Typo: “severe [4]/ The…”

Corrected

Were all the infants who fulfilled the inclusion criteria during the study period included? A flowchart showing the eligible patients and the exclusions with the reason for it would be very interesting to see.

Added following flowchart of patient selection

Results.

Line 128-9. Please avoid “postmenstrual gestational age”.  PMA has been already explained in the abstract and in line 74. It is not necessary to repeat it again. Use just PMA.

Corrected

The same for RSS. It is not necessary to define it again.

Corrected

Figure 2. Please, explain what each of the symbols mean (circles, squares, crosses, etc.).

Corrected

Discussion.

Line 181. It reads “at 26.3 weeks +/- 2.7.5 Sildenafil was started…” Please, correct it.

Corrected

In this paragraph, reference [12] is cited twice (lines 204 and 208), but probably you meant [14]. Please, double check.

Corrected

Round 2

Reviewer 1 Report

.

Author Response

no comments seen

Reviewer 2 Report

Thanks for the effort in the revision. Minor issues have been adequately resolved. However, as explained previously, the main problem of this work is methodological. Since there is no control group, it cannot be concluded that the observed effects are due to the intervention or the result of the natural history of the disease.

Author Response

thank you for your comments. revisions have been made and summarized in letter to the editor